# Determination of Shear Bond Strength between PEEK Composites and Veneering Composites for the Production of Dental Restorations

**DOI:** 10.3390/ma16093286

**Published:** 2023-04-22

**Authors:** Anamarija Kuchler Erjavec, Klementina Pušnik Črešnar, Iztok Švab, Tomaž Vuherer, Majda Žigon, Mihael Brunčko

**Affiliations:** 1Faculty of Polymer Technology, Ozare 19, 2380 Slovenj Gradec, Slovenia; anamarija.kuchler@ftpo.si (A.K.E.);; 2Faculty of Mechanical Engineering, University of Maribor, Smetanova ulica 17, 2000 Maribor, Slovenia; 3Isokon d.o.o., Industrijska cesta 16, 3210 Slovenske Konjice, Slovenia

**Keywords:** BioHPP, CAD/CAM milling, BioHPP plus, pressing, veneering composites, roughness, wettability, shear bond strength

## Abstract

We studied the shear bond strength (SBS) of two PEEK composites (BioHPP, BioHPP plus) with three veneering composites: Visio.lign, SR Nexco and VITA VM LC, depending on the surface treatment: untreated, sandblasted with 110 μm Al_2_O_3_, sandblasted and cleaned ultrasonically in 80% ethanol, with or without adhesive Visio.link, with applied Visio.link and MKZ primer. For the BioHPP plus, differential scanning calorimetry (DSC) revealed a slightly lower glass transition temperature (*T_g_* 150.4 ± 0.4 °C) and higher melting temperature (*T_m_* 339.4 ± 0.6 °C) than those of BioHPP (*T_g_* 151.3 ± 1.3 °C, *T_m_* 338.7 ± 0.2 °C). The dynamical mechanical analysis (DMA) revealed a slightly higher storage modulus of BioHPP (*E’* 4.258 ± 0.093 GPa) than of BioHPP plus (*E*′ 4.193 ± 0.09 GPa). The roughness was the highest for the untreated BioHPP plus, and the lowest for the polished BioHPP. The highest hydrophobicity was achieved on the sandblasted BioHPP plus, whereas the highest hydrophilicity was found on the untreated BioHPP. The highest SBSs were determined for BioHPP and Visio.lign, adhesive Visio.link (26.31 ± 4.17 MPa) or MKZ primer (25.59 ± 3.17 MPa), with VITA VM LC, MKZ primer and Visio.link (25.51 ± 1.94 MPa), and ultrasonically cleaned, with Visio.link (26.28 ± 2.94 MPa). For BioHPP plus, the highest SBS was determined for a sandblasted surface, cleaned ultrasonically, with the SR Nexco and Visio.link (23.39 ± 2.80 MPa).

## 1. Introduction

Polyetheretherketone (PEEK) is a member of the class of high performance polymers (HPP), with the common name polyaryletherketones (PAEK). PEEK is a semi-crystalline, polycyclic, aromatic polymer, where the aromatic rings are bonded with ether and ketone functional groups. The glass transition temperature (*T_g_*) of PEEK is 143 °C and the melting temperature (*T_m_*) is 343 °C [1,2,3,4]. PEEK became an interesting material for biomedical and dental applications because of its favourable features, such as high temperature resistance, biocompatibility and resistance to inorganic and organic chemicals. PEEK has a Young’s modulus (*E*) of 4 GPa, [1,5,6,7,8,9,10,11], which is much lower than that of titanium (102–110 GPa) [9,10], and very close to the human trabecular bone (1 GPa) [10]. The *E* of PEEK can be improved when using PEEK composites, and comes closer to the *E* of cortical bone (14–18 GPa) [1,9,10,12]. Immune reactions (type IV) can appear when using titanium alloys as a result of the possible release of titanium and metallic ions when the protective TiO_2_ layer on titanium alloy implants breaks down as a consequence of the micromotions caused by cyclic loading and the acidic environment in the oral cavity. This is one of the reasons PEEK can also be used as an alternative to the titanium alloys used for artificial bone and orthopaedic spine or dental implants [13]. The limitation of PEEK is its osseointegration, which can be enhanced with a coating of bioactive particles [13,14,15,16]. The technological improvement to processing and surface treatment has enabled the easier and more common use of PEEK in dentistry for fixed and removable prostheses and their components, individual abutments for dental implants, dental implants, orthodontic wires and maxillary obturator prostheses [1,5,6,7,8,9,10,11,12,17,18,19,20,21].

PEEK can also be used as a composite, for example, with carbon or titanium dioxide (TiO_2_) fillers [22]. TiO_2_ is known for its high antibacterial activity, which is why it is used in dentistry [22]. PEEK composites have better mechanical properties than PEEK, they can reach a Young’s modulus of about 15–18 GPa, and are also used in dentistry [1,9,21,23,24,25]. However, PEEK composites alone are not suitable for aesthetic fixed or removable restorations due to their monotonous color. This is why we use them as substructures for prosthetic work such as crowns and bridges. To achieve the desired aesthetic appearance, we need to apply veneering dental composites. The hydrophobic surface of PEEK has to be pretreated before veneering to increase its hydrophilicity, which is necessary for achieving an adequate bond strength with the veneering composite. One of the most efficient methods of surface pretreatment, in addition to etching with sulfuric acid, according to the literature, is sandblasting with aluminium oxide (Al_2_O_3_) and the use of adhesives based on methyl methacrylate (MMA) and dimethacrylates (DMA) [3,5,8,9,12,17,18,23,24,26,27,28,29,30,31,32,33,34,35,36,37,38,39,40,41,42].

The shear and tensile strengths between PEEK, reinforced with TiO_2_ or SiO_2_, and veneering composites or dental cements have been the subject of numerous studies [9,28,30,34,43,44]. The shear bond strength (SBS) has been investigated after various surface pretreatments (untreated, sandblasted with 50 μm Al_2_O_3_, silica coating or laser treatment etc., and the use of different adhesives or primers). Different veneering composites have been applied: Crea.lign, SR Nexco and different cements. Some authors measured the surface roughness after different surface pretreatments [8,9,28,30,34,41,43,44].

In our literature search, we did not come across any comparison of the SBS between BioHPP and BioHPP plus, both with good mechanical properties and a suitable color, with different veneering composites. The purpose of this study was, therefore, to determine the type of PEEK composite (BioHPP or BioHPP plus, the latter having a higher proportion of filler), selected surface treatment (no treatment, sandblasted with 110 μm Al_2_O_3_, sandblasted and cleaned ultrasonically with 80% ethanol, sandblasted with the application of MKZ primer, sandblasted and applied with MKZ primer and adhesive Visio.link, sandblasted and applied adhesive Visio.link, and sandblasted, cleaned ultrasonically with 80% ethanol, with applied adhesive Visio.link) and type of veneering composite that will enable the highest possible SBS.

## 2. Materials and Methods

### 2.1. Used Materials

#### 2.1.1. BioHPP

In dentistry, a PEEK composite with 20 wt% TiO_2_ filler is used for abutments in dental implants, temporary abutments and frameworks for fixed or removable dental restorations. The term for a PEEK composite with 20 wt% TiO_2_ is Biocompatible High Performance Polymer (BioHPP, REF: 54002030). A BioHPP with a semi-crystalline structure belongs to the group of technical polymers. The TiO_2_ filler, in the form of fibres with a length of 0.3–0.5 μm, enables better mechanical properties than unfilled PEEK. The Young’s modulus *E* of BioHPP is similar to the Young’s modulus of PEEK. The material is on the market in a form for milling (the CAD/CAM technique) or in pellets/granulate for pressing [6,7,11,12,23,24,25,34,43,44,45].

#### 2.1.2. BioHPP Plus

A newer type of PEEK composite (BioHPP plus) is also on the market, and has, according to the manufacturer (Bredent GmbH, Senden, Germany, REF: 54F2PP15), a higher Young’s modulus than BioHPP. The higher modulus should positively influence the shear strength with veneering composites. BioHPP plus consists of 24 wt% inorganic fillers, for example TiO_2_ and 1 wt% of an inorganic pigment. Pressed BioHPP plus specimens were manufactured using a For2press unit (Bredent GmbH). 

#### 2.1.3. Veneering Composites: Visio.lign, SR Nexco and VITA VM LC

We used three veneering composites that differ in their polymer matrix and filler content: Visio.lign, SR Nexco and VITA VM LC (Table 1). According to the recommendation of the manufacturer of both PEEK composites, the Visio.lign composite system is the veneering composite of choice for BioHPP and BioHPP plus. In addition, veneering composite VITA VM LC with EDMA and TEGDMA matrix showed better tensile bond strengths for unfilled PEEK compared to an UDMA and EDMA veneering composite (GC Gradia) when the specimens were sandblasted with 50 μm Al_2_O_3_ and adhesive Visio.link was applied. The results were lower when plasma was used as the surface treatment method of unfilled PEEK and PEEK composites with 20% TiO_2_. The results were higher for the PEEK composites with 20% TiO_2_ and 1% pigment powder [5,39].

### 2.2. Test Specimens and Characterisation

#### 2.2.1. BioHPP and BioHPP Plus Test Specimens for Differential Scanning Calorimetry (DSC) Analysis

DSC analysis was performed on two specimens of both composites. The mass of the specimens was 8 mg.

#### 2.2.2. BioHPP and BioHPP Plus Test Specimens for Dynamical Mechanical Analysis (DMA) Analysis

For the DMA analysis, test specimens were manufactured in the shape of a rectangular solid, 5 mm × 2 mm × 45 mm in size. The BioHPP specimens were milled using a CAD/CAM milling machine, InLab MC X5, Dentsply Sirona Inc (Bensheim, Germany). We used different dental burs (carbide dental burs with a diameter of 2.5 mm, REF H200 M823, REF H194 M8 40 and REF H272 M8 14) to remove excess material and irregularities at the surface. The pressed specimens were manufactured according to the manufacturer’s instructions. The cylinder-shaped specimens were 3D printed with a photopolymer resin (FotoDent cast resin) using Asiga Max UV, Puretone Ltd (Sydney, Australia). The 3D printed specimens were used to make a silicone mould, into which we poured dental wax (Thowax modelierwachs, beige, Yeti, LOT: 01111017). After pressing the specimens, they were sandblasted with 110 μm Al_2_O_3_ (Chromokorund 110 μm, REF: 093904) and all excess material was removed.

#### 2.2.3. BioHPP and BioHPP Plus Test Specimens for the SBS Test, Wettability and Surface Roughness Measurements

Overall, 273 specimens of both PEEK composites were manufactured for this study, in a cylindrical shape, with a diameter of 4.3 mm and height of 1 cm (Figure 1). The surface treatments are listed in Table 2. The specimens were arranged according to the framework material and surface treatment used (Table 3). The first series, containing 147 BioHPP specimens, was milled out of blanks with the CAD/CAM milling machine. The second series, comprising 126 BioHPP plus specimens, was pressed out of pellets. The manufacturing of the BioHPP and BioHPP plus test specimens is described in Section 2.2.2.

The BioHPP specimens were classified into seven groups, each comprising 21 specimens, according to the surface treatment, whereas the BioHPP plus specimens were classified into six groups (we did not include the untreated group) (Table 3).

Sandblasting was performed according to the manufacturer’s instructions with 110 μm Al_2_O_3_, for 5 s, under a 45° angle and a pressure of 2.5 bar (groups B, C, D, E, F and G). Adhesive Visio.link (REF: VLPMMA10) and MKZ primer (REF: MKZ02004) were used according to the manufacturer’s instructions (groups D, E, F and G). Groups C and G were cleaned in an ultrasonic cleaner with 80% ethanol (Finevo 01.2 ethanol 80%, REF: 507157), 5 min at 30 °C.

The groups were divided according to the surface treatments of BioHPP and BioHPP plus (mentioned further in the groups as the mark +) and the veneering composite used (Table 3):Group A (untreated): with the veneering composite Visio.lign (AV for BioHPP), with the veneering composite SR Nexco (AN for BioHPP) and the veneering composite VITA VM LC (AVI for BioHPP);Group B (sandblasted with 110 μm Al_2_O_3_): with the veneering composite Visio.lign (BV for BioHPP and BV+ for BioHPP plus), with the veneering composite SR Nexco (BN for BioHPP and BN+ for BioHPP plus) and the veneering composite VITA VM LC (BVI for BioHPP and BVI+ for BioHPP plus);Group C (sandblasted with 110 μm Al_2_O_3_, cleaned ultrasonically with 80% ethanol, 5 min): with the veneering composite Visio.lign (CV for BioHPP and CV+ for BioHPP plus), with the veneering composite SR Nexco (CN for BioHPP and CN+ for BioHPP plus) and the veneering composite VITA VM LC (CVI for BioHPP and CVI+ for BioHPP plus);Group D (sandblasted with 110 μm Al_2_O_3_, application of MKZ primer): with the veneering composite Visio.lign (DV for BioHPP and DV+ for BioHPP plus), with the veneering composite SR; Nexco (DN for BioHPP and DN+ for BioHPP plus) and the veneering composite VITA VM LC (DVI for BioHPP and DVI+ for BioHPP plus);Group E (sandblasted with 110 μm Al_2_O_3_, application of MKZ primer and adhesive Visio.link): with the veneering composite Visio.lign (EV for BioHPP, EV+ for BioHPP plus), with the veneering composite SR Nexco (EN for BioHPP and EN+ for BioHPP plus) and with VITA VM LC (EVI for BioHPP and EVI+ for BioHPP plus);Group F (sandblasted with 110 μm Al_2_O_3_, application of adhesive Visio.link): with the veneering composite Visio.lign (FV for BioHPP and FV+ for BioHPP plus), with the veneering composite SR Nexco (FN for BioHPP and FN+ for BioHPP plus) and with the veneering composite VITA VM LC (FVI for BioHPP and FVI+ for BioHPP plus);Group G (sandblasted with 110 μm Al_2_O_3_, cleaned ultrasonically with 80% ethanol, 5 min, application of adhesive Visio.link): with the veneering composite Visio.lign (GV for BioHPP and GV+ for BioHPP plus), with the veneering composite SR Nexco (GN for BioHPP and GN+ for BioHPP plus) and with the veneering composite VITA VM LC (GVI for BioHPP and GVI+ for BioHPP plus).

We also manufactured nine BioHPP and six BioHPP plus cylinder-shaped specimens for the wettability and roughness measurements. The nine BioHPP specimens were divided into three groups: untreated, sandblasted with 110 μm Al_2_O_3_ and polished specimens. As untreated BioHPP plus specimens, we used specimens from which we had removed all casting material. Polishing of both PEEK composites was performed after removing all irregularities, first with rough burs, then with a polishing rubber disc, and last with Acrypole, REF 520 001 70, and a white goat hairbrush, REF 350 005 40. For high gloss polishing, we used Abraso-Starglanz, REF 520 001 63, and a cotton buff, REF 350 006 50.

### 2.3. DSC Analysis

Thermal characterisation (the glass-transition temperature, *T_g_*, the melting temperature, *T_m_*, and the crystallisation temperature, *T_c_*) was carried out using a differential scanning calorimeter, Mettler Toledo DSC 2 Star System. The specimens were dried before the analysis at 100 °C until all the water evaporated, using a Mettler Toledo HX024 moisture analyser. They were heated, subsequently cooled, and heated again in the temperature range of between 50 °C and 380 °C. The heating and cooling rates were 10 °C min^−^^1^. All analyses were performed under a nitrogen atmosphere.

The degree of crystallinity (*X_c_*) of both PEEK composites was calculated from the enthalpy of fusion (Equation (1)) [46]:(1)$$X_{c}\left(\%\right)=\frac{\Delta H_{f}}{\Delta H_{f,c}}\cdot \frac{1}{w}\cdot 100$$
where: Δ*H_f_*—is the enthalpy of fusion, Δ*H_f,c_*—is the enthalpy of fusion of the fully crystalline PEEK (130 Jg^−^^1^) [2], *w*—is the mass fraction of polymer in the PEEK composite

### 2.4. DMA Analysis

The viscoelastic properties (storage modulus, *E′*, and loss modulus, *E″*, damping factor, tan δ) and the glass transition temperature (*T_g_*) of both PEEK composites were determined using a dynamical mechanical analyser DMA 8000, Perkin Elmer, with a dual cantilever, in accordance with ASTM D7028-07 (2015). The DMA was performed at a frequency of 1 Hz, an amplitude of 0.01 mm and a heating rate of 2 °C min^−^^1^ in the temperature range 30 °C–250 °C.

### 2.5. Wettability and Roughness Measurements

The wettability and roughness measurements were performed on untreated specimens, specimens sandblasted with 110 μm Al_2_O_3_ and polished specimens. For the wettability measurements, we used a goniometer from DataPhysics, Goniometer OCA 35 (Filderstadt, Germany). Two test liquids were used: polar liquid (ultra-pure water, Millipore, Burlington, MA, USA) and nonpolar liquid (diiodomethane, Sigma-Aldrich, Burlington, MA, USA 99%), with a drop volume of 3 μL. The roughness measurements were performed using a contact profilometer, Mitutoyo SJ-210, to determine the mean roughness value (*R_a_*) of the specimens at a scanning rate 0.25 mm s^−^^1^, measuring track 2 mm and a distance between the tracks of 0.08 mm.

The surface free energies were calculated according to the Owen, Wendt, Rabel, Kaelble (OWRK) method (Equation (2)) using the computer program SCA-20 [47,48]:(2)$$\gamma_{lv}\left(1+cos\;\theta \right)=2\sqrt{\gamma_{s}^{d}\gamma_{l}^{d}}+2\sqrt{\gamma_{s}^{p}\gamma_{l}^{p}}$$
where *γ_lv_*—is the surface energy of the liquid, *θ*—is the contact angle between the test liquid and tested material, *γ_s_^d^*—is the disperse part of the surface free energy of the solid, *γ_l_^d^*—is the disperse part of the surface free energy of the test liquid, *γ_s_^p^*—is the polar part of the surface free energy of the solid and *γ_l_^p^*—is the polar part of the surface free energy of the test liquid.

### 2.6. Shear Bond Strength (SBS) Test

The loading of the specimens under shear stress is clinically more relevant than flexural or tensile stress, as shear stress occurs more commonly during chewing. In our case, we measured the shear strength between the framework material (BioHPP, BioHPP plus) and the three veneering composites with a crosshead speed of 0.01 m s^−^^1^ [12,17,23].

The SBS test was performed on a universal testing machine for the mechanical test, a Smitweld simulator–Tensile Test Unit 2002 (Nijmegen, The Netherlands), according to the standard ISO 10477: 2004 (E). A special mould, made of two parts and a hole through both parts, was inserted into the machine. The specimens were placed in the hole with a diameter of 4.3 mm and length of 1 cm. The interface between the substructure and veneering composite was in the middle of both mould parts.

### 2.7. Comparison of Stereomicroscope Images of the Interface Layer between the PEEK and Veneering Composites after the SBS Test

The analysis of the fracture surfaces was performed on a stereomicroscope Olympus SZX10 (Tokyo, Japan). For the analysis, we selected one random specimen from six groups: the specimens of the two groups that achieved the lowest SBS (AN and DVI+) and the specimens that achieved the highest SBS for BioHPP (group FV) were compared to group FV+ with the same surface treatment. In addition, group GN+, which achieved the highest SBS for BioHPP plus, was compared with group GN. We defined whether the failure was adhesive, cohesive or mixed. Adhesive failure, or delamination, is a failure along the interface surface between the PEEK framework and veneering composite resin, whereas cohesive failure occurs in the adhesive layer itself; a layer of adhesive remains on both surfaces, that is, the PEEK and the veneering composite resin. In a mixed failure, the adhesive and cohesive failure modes occur simultaneously [49,50].

## 3. Results

This section provides a summary of the DSC analysis, DMA analysis and the roughness, wettability and SBS measurements.

### 3.1. DSC Analysis

The glass transition temperatures of both PEEK composites differed by 0.9 °C, whereas the differences with the manufacturer’s values were greater (Table 4, curves available in Appendix A). The *T_g_* of BioHPP (151.3 °C) was lower than the *T_g_* given by the manufacturer (155.8 °C), but higher than the *T_g_* of BioHPP plus (150.4 °C). This could be a consequence of the different manufacturing methods (CAD/CAM and pressing) and/or the composition of the PEEK composites. In addition, the DMA analysis showed a higher *T_g_* for BioHPP (169 °C) than for BioHPP plus (167 °C). However, the DMA and DSC values of *T_g_* differed because of the diverse measurement methods of both analyses [51]. The melting point, *T_m_*, of BioHPP (338.7 °C) was lower than the *T_m_* of BioHPP plus (339.4 °C) and the *T_m_* according to the manufacturer (345.1 °C). The degree of crystallinity (*X_c_*) of BioHPP plus (27.9%) was lower than the degree of crystallinity of BioHPP (29.0%), and may affect the reduction in the *T_g_* value of BioHPP plus [20,52,53]. The first heating curve of BioHPP plus showed a cold crystallisation (*T_cc_*) at 170.2 °C, which occurs simultaneously during the heating of quenched PEEK [54]. The temperature of crystallisation (*T_c_*) of BioHPP (287.6 °C) could be lower than the *T_c_* of BioHPP plus (293.0 °C) because of the increased filler content in BioHPP plus (24 wt% inorganic filler and 1 wt% inorganic pigment) [55].

### 3.2. DMA Analysis

The viscoelastic properties of BioHPP and BioHPP plus were determined through DMA (Table 5). The storage modulus (*E*′) of BioHPP is higher than the *E*′ of BioHPP plus. The results show that BioHPP plus is less rigid than BioHPP, despite the higher proportion of fillers, which may also be a consequence of the lower degree of crystallinity of BioHPP plus [54]. The loss modulus (*E*″) of BioHPP is lower than the *E*″ of BioHPP plus. The damping factors (tan δ) of BioHPP and BioHPP plus are the same, within a standard deviation value. The differences in the viscoelastic properties could be a consequence of the different manufacturing procedures, leading to some structural inhomogeneities that can occur during pressing, casting, heating and holding at a certain temperature, etc. [4,56].

### 3.3. Wettability and Roughness

The wettability and roughness tests were measured on the BioHPP and BioHPP plus specimens with various surface treatments (untreated, sandblasted specimens with 110 μm Al_2_O_3_ and polished specimens). Table 6 represents the results of the wettability of BioHPP in water. The water contact angle of the BioHPP surface was θ > 90°, except for the untreated and sandblasted surfaces, indicating a hydrophobic nature. On the contrary, the lowest contact angles and the most hydrophilic surfaces were those on the untreated BioHPP specimens, where the water contact angle was θ < 90°. For all of the samples, the changes in the contact angle depend on the treatment of the specimen’s surface, and were comparable to the results in the literature [26,30,37]. In addition, the changes in the BioHPP and BioHPP plus composites were evident.

The surface free energy of the BioHPP and BioHPP plus specimens (Table 7) was calculated using the OWRK approach [47]. The results, in Table 7, show the main differences in the surface free energy, depending on the surface treatment of the BioHPP and BioHPP plus composites; the surfaces were either untreated, sandblasted or polished. In addition, sandblasting influenced the surface free energy of both BioHPP and BioHPP plus. Thus, it was calculated that the highest surface free energies were those on the sandblasted BioHPP plus specimens. When both the PEEK composites were polished, the specimens showed lower surface free energy than when untreated and sandblasted. The difference in the surface free energy of BioHPP and BioHPP plus could be the result of the different processing methods (CAD/CAM and pressing) and different surface geometries of the BioHPP and BioHPP plus specimens [10,57].

The results in Table 8 show the average roughness of BioHPP and BioHPP plus. The BioHPP specimens showed the lowest roughness.

More specifically, the lowest roughness of the BioHPP specimens was measured on the polished surface; on the other hand, the highest roughness of the BioHPP was measured on the sandblasted specimens (Table 8), which agrees with the results reported in the literature [17,23,26,28,31,34,43]. In the case of BioHPP plus, the highest roughness was characteristic of an untreated surface, and the lowest roughness of a polished surface, comparable to BioHPP.

### 3.4. Shear Bond Strength (SBS)

In general, the BioHPP plus specimens showed lower SBS in comparison to the BioHPP specimens (Table 9 and Figure 2). The lowest SBS results of all the specimen groups are those of the BioHPP specimens, group A, with untreated surfaces. The highest SBSs were observed for the following BioHPP specimens: (a) Visio.lign veneering composite system after sandblasting and application of adhesive Visio.link (26.31 MPa); (b) VITA VM LC veneering composite, cleaned ultrasonically in 80% ethanol before the adhesive Visio.link was applied (26.28 MPa). The BioHPP specimens veneered with VITA VM LC also showed a high SBS, with: (a) MKZ primer (25.59 MPa); (b) MKZ primer and Visio.link adhesive (25.51 MPa). For BioHPP plus, the highest SBS was determined for the sandblasted surface, cleaned ultrasonically in 80% ethanol and with the veneering composite SR Nexco (23.39 MPa).

Two groups (AN and DVI+, Table 3 and Table 9) did not reach the minimum required SBS of 5 MPa in the oral cavity, according to ISO 10,477 [23,28,43]. SBS higher than 10 MPa for PEEK used in veneering composite has been reported as clinically acceptable [8,26,29,35,39] and was not achieved in groups AV, AN, AVI, BN, CVI+, DN, DN+, DVI+ and EVI+. All the specimens in group D with a MKZ primer, with the exception of group DV, showed SBSs below 15 or 10 MPa. BioHPP with the veneering composite SR Nexco, as well as BioHPP plus with all the veneering composites, showed lower SBS with the use of Visio.link adhesive and a MKZ primer (group E) than specimens where only Visio.link adhesive was applied (group F). However, EV+ and EN+ showed SBSs higher than 10 MPa (15.70 MPa; 14.34 MPa). We believe that the lower SBS with the use of MKZ primer may imply impurities on the surfaces because we waited 30 s for the primer to dry. The specimens of group G, which were cleaned ultrasonically with 80% ethanol after sandblasting and then Visio.link adhesive was applied, showed a lower SBS on BioHPP with the Visio.lign composite system (20.31 MPa) compared to the BioHPP specimens of group G with the veneering composites SR Nexco (25.00 MPa) and VITA VM LC (26.28 MPa). However, the SBSs of all the G groups were much higher than the clinically acceptable shear bond strength (10 MPa). The lower SBS of the BioHPP plus in group G was achieved only with the use of the veneering composite VITA VM LC. Group C (sandblasting and ultrasonic cleaning with 80% ethanol) showed better SBS than group B (only sandblasting), the only exceptions being the BioHPP plus specimens with VITA VM LC.

The results show that the percentage value of standard deviations was the lowest in group EVI (7.6%), whereas the highest percentage value of standard deviations was in group AN (111%). When comparing the percentage of standard deviations for each group of veneering composites on BioHPP, the highest percentage was in the group A without surface preparation (AV, 53.1%; AN, 111%; AVI, 58.8%). This could be a consequence of the surface roughness and the surface that had the highest percent of standard deviation (45.8%). A higher percentage of standard deviations was seen also in groups DN (84.4%), BN (60.4%), EN (41.6%) and BVI (41.6%). This could also be a consequence of the surface roughness, due to irregularities on the surface.

The shear modulus was calculated after the SBS test (Figure 3). We can conclude that the shear modulus results were quite in agreement with the SBS test results. The highest shear modulus was achieved by groups DV, EV and FV. Groups EVI and FVI, with two of the highest SBSs, had lower shear modulus results than groups DV and EV, but were still quite high compared to the other groups.

### 3.5. Optical Analysis of Fractured Surfaces with a Stereomicroscope

Figure 4 shows the stereomicroscope images of the fractured surfaces after the SBS test. From these images, we concluded whether the failure was adhesive, cohesive or mixed. None of the failures were completely cohesive, as only adhesive and mixed failures were observed (Table 10). Adhesive failures were observed for groups AN, DVI+ and FV+, whereas mixed failures occurred in groups GN, GN+ and FV. Group DVI+ also showed minor porosities, which occurred during veneering.

## 4. Discussion

The SBS of the PEEK composites BioHPP and BioHPP plus with different veneering composites is influenced by many factors, such as the surface roughness and wettability, type of surface pretreatment and the use of adhesives and/or primers. Many studies have reported on the surface roughness, wettability and SBS of BioHPP with different veneering composites and cements, whereas, to the best of our knowledge, studies on BioHPP plus have not previously been conducted [23,26,28,30,34,43].

The contact angle with distilled water (Table 6) for untreated BioHPP (75.57°) was similar to the results of BioHPP reported in the literature (79.67°) [30]. The contact angle of sandblasted BioHPP was 89.75°, which was higher than reported (84.83°) [30]. The contact angle with distilled water of sandblasted BioHPP plus (106.87°) was higher than of sandblasted BioHPP (89.75°), implying the higher hydrophilicity of sandblasted BioHPP. The surface free energy of sandblasted BioHPP plus was higher than that of BioHPP, which could be a consequence of the higher roughness of BioHPP plus. The polished specimens of both PEEK composites showed low surface free energy. However, the low surface free energy of polished specimens is desired because it decreases the possibility of bacteria adhering to the surface [58]. The untreated surfaces of BioHPP showed lower surface free energies than the unfilled PEEK (48.4 mN/m) [17], whereas BioHPP plus showed higher surface free energies than the unfilled PEEK, which indicates that the filler influences the surface free energies of both PEEK composites. Surface sandblasting increases the contact angle with distilled water. Nevertheless, our goal was to decrease the contact angle with distilled water and increase the surface hydrophilicity, which would positively influence the bond strength with veneering composites.

BioHPP plus also follows the trend of an increased contact angle with distilled water after sandblasting, although the roughness of the sandblasted surfaces was lower than that of the untreated BioHPP plus. The contact angle of the polished BioHPP plus surfaces was slightly lower, and the hydrophilicity was higher than that of the polished BioHPP surfaces. The polished BioHPP and BioHPP plus showed similar results in their surface free energies (40.93 mN/m and 38.80 mN/m, Table 7) to those reported in the literature (44.9 mN/m) [58].

The surface roughness of the BioHPP specimens differed depending on the type of pretreatment (Table 8): the highest surface roughness was found for sandblasted surfaces; those of the untreated BioHPP specimens were lower than the previous ones, whereas the lowest surface roughness was characteristic of the polished specimens. The surface roughness of the untreated BioHPP (0.568 μm, Table 8) was lower or higher than the values reported: 1.03 μm [28], 1.11 μm [43], 0.33 μm [34], 0.03 μm [26], 3.32 μm [23]. This could be a consequence of using different analysing methods, different measurement conditions or different surface treatments. Our results for the sandblasted specimens using 110 μm Al_2_O_3_ were lower (1.154 μm) than those reported: 2.26 μm [30] and 6.58 μm [23]. The surface roughness of the polished BioHPP specimens (0.014 μm) was comparable to the results of the manufacturer (0.0239 μm), and the results of both BioHPP and BioHPP plus were lower than the results of pressed BioHPP reported in the literature (0.034 μm), which could be a consequence of different test parameters being used (the profile length was 1.75 mm and resolution 0.01 μm) [58].

The surface roughness of all the BioHPP plus groups (untreated, sandblasted and polished specimens) was higher than that of BioHPP. The surface roughness of the BioHPP plus sandblasted with 110 μm Al_2_O_3_ was lower than that of the untreated specimens, which was the opposite to BioHPP. The lowest surface roughness was characteristic of polished surfaces, the same as for BioHPP. The slightly higher surface roughness of the polished BioHPP plus specimens compared to BioHPP could be a consequence of the different methods of manufacturing. During the compression of BioHPP plus specimens, some minor porosities can evolve, which is not the case for CAD/CAM BioHPP blanks [23,26,28,30,34,43].

The SBS of the untreated BioHPP specimens with a Visio.lign composite system in our study (7.27 MPa, Table 9 and Figure 2) was higher than that reported in the literature [23,28,59], where they did not use Combo.lign preopaquer (6.35 MPa; 5.09 MPa; 5.68 MPa). These results imply that the use of Combo.lign preopaquer is necessary to achieve higher SBS. Higher results with the Visio.lign veneering composite system, as in our study, were also achieved on untreated BioHPP with the use of Visio.link adhesive, Crea.lign opaquer and Crea.lign veneering composite after the specimens were aged in distilled water at 37 °C for 24 h (7.7 MPa) [27].

The results of the SBS between the BioHPP and Visio.lign composite system, sandblasted with 110 µm Al_2_O_3_ and with applied Visio.link adhesive (26.31 MPa) (Table 9 and Figure 2), were similar to, and higher than, the ones in the literature [23,28,30,34,59]. In one study [59], BioHPP specimens were milled and sandblasted with 50 µm Al_2_O_3_ (24.71 MPa), whereas, in another study [30], the specimens were sandblasted with 110 µm Al_2_O_3_ (10.81 MPa). Visio.link adhesive and Combo.lign cement were applied in both studies. In the study [23], the specimens were sandblasted with 110 µm Al_2_O_3,_ and Visio.link adhesive and a Crea.lign veneering composite were applied (12.85 MPa). In the study [34], BioHPP specimens were milled and sandblasted with 110 µm Al_2_O_3_ (6.14 MPa), whereas in another study [28], the specimens were sandblasted with 50 µm Al_2_O_3_, and then the Visio.link adhesive, Crea.lign opaquer and Crea.lign veneering composite were applied (10.97 MPa). In all cases, the results of the SBS were lower than the BioHPP in our study. The SBS results of the BioHPP plus after sandblasting with 110 µm Al_2_O_3_ and the application of Visio.link adhesive (19.62 MPa) were lower than those of the BioHPP with the same surface treatment [59] (24.71 MPa), or higher [34] (6.14 MPa; 10.81 MPa).

The SBSs of BioHPP and BioHPP plus after sandblasting with 110 µm Al_2_O_3_ and the application of Visio.link adhesive and SR Nexco veneering composite were 19.76 MPa (BioHPP) and 18.44 MPa (BioHPP plus, Table 9). In comparison to the literature, our results were lower than those of milled BioHPP specimens after the same surface treatment, which were aged for 24 h (31.1 MPa) [40] or higher (13.60 MPa) [34]. The difference in results may be due to the fact that our specimens were not subjected to ageing.

In general, the SBSs of the BioHPP plus specimens were lower than those of BioHPP; this could be the result of the thermal pretreatment during pressing, which did not occur during the manufacturing of the BioHPP discs. In the study [60], they tested the influence of thermal pretreatment during the fused deposition modelling (FDM) of carbon-reinforced PEEK (CRF-PEEK) on the degree of crystallinity. They found that the degree of crystallinity increased with higher post-processing temperatures. Thus, we can assume that the degree of crystallinity of BioHPP plus is affected by lower post-processing thermal treatment and can lead to a reduction in the tensile strength and other mechanical properties. It is known that the higher the degree of crystallinity, the higher the tensile strength and mechanical properties of the material [60]. The highest results of SBS in our study were achieved for BioHPP sandblasted with 110 μm Al_2_O_3_, with the veneering composite Visio.lign and adhesive Visio.link (26.31 MPa), as well as for BioHPP sandblasted with 110 μm Al_2_O_3_, cleaned ultrasonically with 80% ethanol, with the veneering composite VITA VM LC and adhesive Visio.link (26.28 MPa).

Other surface treatments studied in the literature are: acid etching with sulfuric acid or piranha solution and plasma treatment or laser treatment [27,28,30,34]. In the studies [30,34], BioHPP specimens were treated with a sulfuric acid solution, and they achieved the highest SBS after artificial ageing (15.82 MPa; 13.80 MPa). Other pretreatments (piranha solution, plasma pretreatment and laser treatment) led to lower SBS than pretreatment with sulfuric acid. The SBS values of BioHPP in our study with the Visio.lign composite system were higher after almost all kinds of pretreatments, except for group AV, whereas, for BioHPP plus, the SBS results were higher only for groups FV+ and GV+. By using the veneering composite SR Nexco, higher SBSs were achieved in groups EN, FN, GN, FN+ and GN+, whereas the VITA VM LC veneering composite showed higher results only in groups EVI, FVI and GVI. The SBSs for groups F and G in our study were higher than the SBSs after pretreatment with sulfuric acid (15.82 MPa; 13.80 MPa, [30,34]), only FVI+ and GVI+ showed lower SBS (15.30; 14.13 MPa). The results in our study confirm the improvement of the SBS between PEEK composites and veneering composites by using Visio.link adhesive on sandblasted surfaces.

The fracture of specimens (Figure 4) was either adhesive or mixed (Table 10). When comparing the stereomicroscope images of groups FV and FV+, we can see that a specimen of group FV had a more cohesive failure than group FV+, which means that the bond between the BioHPP and Visio.lign veneering composite with the use of Visio.link adhesive after sandblasting was established better than for BioHPP plus (group FV+). This could be a consequence of the higher hydrophilicity of sandblasted BioHPP, which ensures good wettability and a better bond with the adhesive Visio.link. Group GN+ achieved the highest SBS for the BioHPP plus specimens, but lower than group GN with BioHPP. As seen on the stereomicroscope image, the fractures of both the GN+ and GN groups were mixed, but with a higher proportion of cohesive failure for the GN specimens. The specimens of groups AN and DV+ showed an adhesive failure, which was expected at low SBSs.

## 5. Conclusions

We studied two different PEEK composites: milled BioHPP (20 wt% filler TiO_2_) and pressed BioHPP plus (24 wt% inorganic fillers and 1 wt% pigment), to determine their viscoelastic properties, surface roughness, wettability and shear bond strength (SBS) after different surface pretreatments with three veneering composites: Visio.lign, SR Nexco and VITA VM LC. Although it was expected that the composite BioHPP plus, with a higher proportion of fillers, would achieve a greater SBS than composite BioHPP, our results did not confirm this hypothesis. We infer that the composition and proportion of the fillers in BioHPP plus may affect the reduction in its degree of crystallinity, and, consequently, also the reduction in SBS with veneering composites.

The highest surface roughness among the BioHPP and BioHPP plus specimens was measured on the untreated BioHPP plus specimens, and the lowest surface roughness on the polished BioHPP specimens. The contact angle with distilled water of BioHPP plus decreased on polished surfaces, leading to increased hydrophilicity. The surface hydrophobicity of BioHPP increased in the following order: untreated surface, sandblasted surface and polished surface. BioHPP plus also showed a higher contact angle with distilled water after surface sandblasting, whereas the contact angle on polished surfaces was slightly lower than that of BioHPP.

The highest SBS of the BioHPP specimens was achieved for specimens whose surface was sandblasted with 110 μm Al_2_O_3_, coated with Visio.link adhesive and then a Visio.lign composite system was applied (group FV, 26.31 MPa). High SBSs were determined for the sandblasted specimens with the composite system Visio.lign, where the MKZ primer was applied (group DV, 25.59 MPa); sandblasted specimens veneered with VITA VM LC with applied MKZ primer and adhesive Visio.link (group EVI, 25.51 MPa); and sandblasted specimens cleaned ultrasonically in 80% ethanol, veneered with VITA VM LC and with applied Visio.link (group GVI, 26.28 MPa). A high SBS was also achieved for BioHPP specimens sandblasted with the SR Nexco veneering composite, where adhesive Visio.link was applied (group GN, 25.00 MPa).

For BioHPP plus, the highest SBSs were determined for the sandblasted surface, cleaned ultrasonically in 80% ethanol: with the veneering composite SR Nexco and Visio.link adhesive (group GN+, 23.39 MPa); and with the veneering composite Visio.lign and Visio.link adhesive (group GV+, 21.53 MPa).

In general, higher SBSs were observed for BioHPP and not for BioHPP plus composites. We can conclude that the higher roughness of the BioHPP plus surface decreases the SBS with veneering composites.

A limitation of this study was not performing long-term water (artificial saliva) storage or thermocycling. Our study had an in vitro design, which is why we did not include the aforementioned tests, and the obtained results should be interpreted by taking into account the limitations of in vitro studies.

## Figures and Tables

**Figure 1 materials-16-03286-f001:**
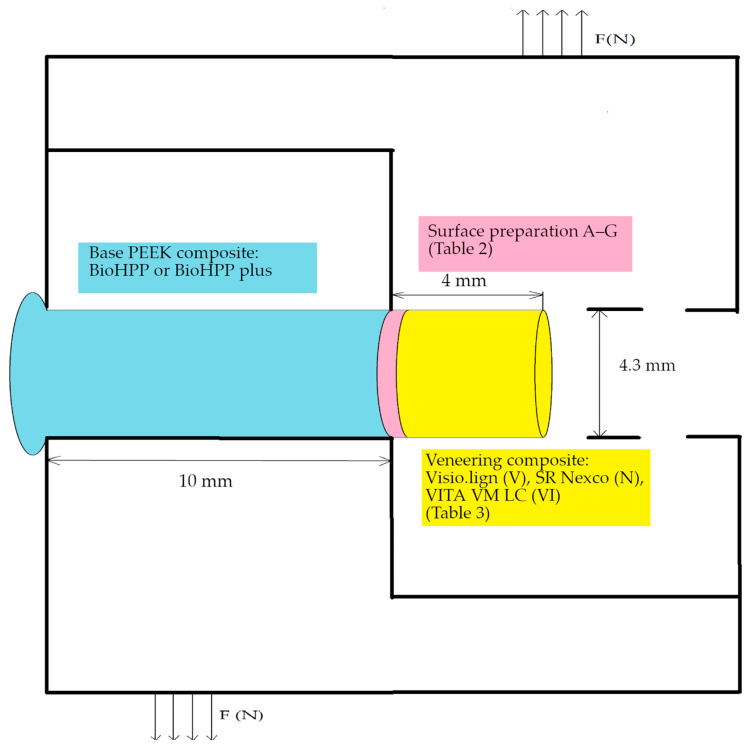
Shear bond strength (SBS) test specimen in the special mould used for the universal testing machine Swithweld simulator–Tensile Test Unit 2002.

**Figure 2 materials-16-03286-f002:**
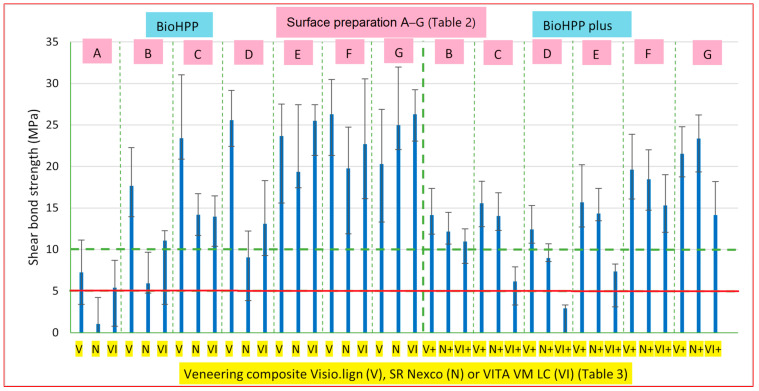
Shear bond strength (SBS) of BioHPP and BioHPP plus specimens with various surface treatments and three different veneering composites: Visio.lign (V), SR Nexco (N) and VITA VM LC (VI). The red line at 5 MPa represents the minimal requested SBS in the oral cavity, and the green line at 10 MPa represents the clinically acceptable SBS between the PEEK and the veneering composite).

**Figure 3 materials-16-03286-f003:**
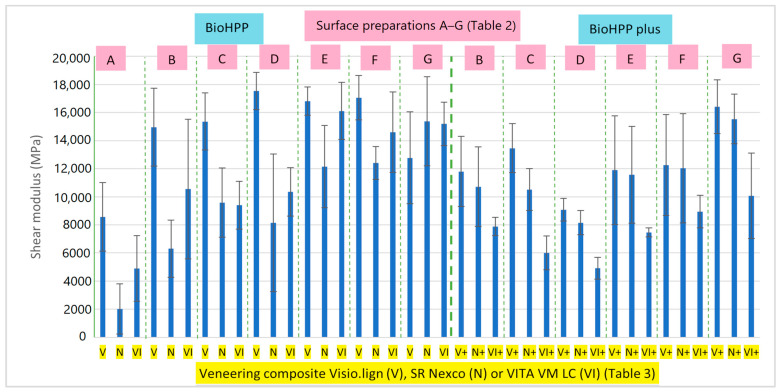
Shear modulus medians and standard deviations calculated after a shear bond strength test of BioHPP and BioHPP plus specimens with various surface treatments and three different veneering composites: Visio.lign (V), SR Nexco (N) and VITA VM LC (VI).

**Figure 4 materials-16-03286-f004:**
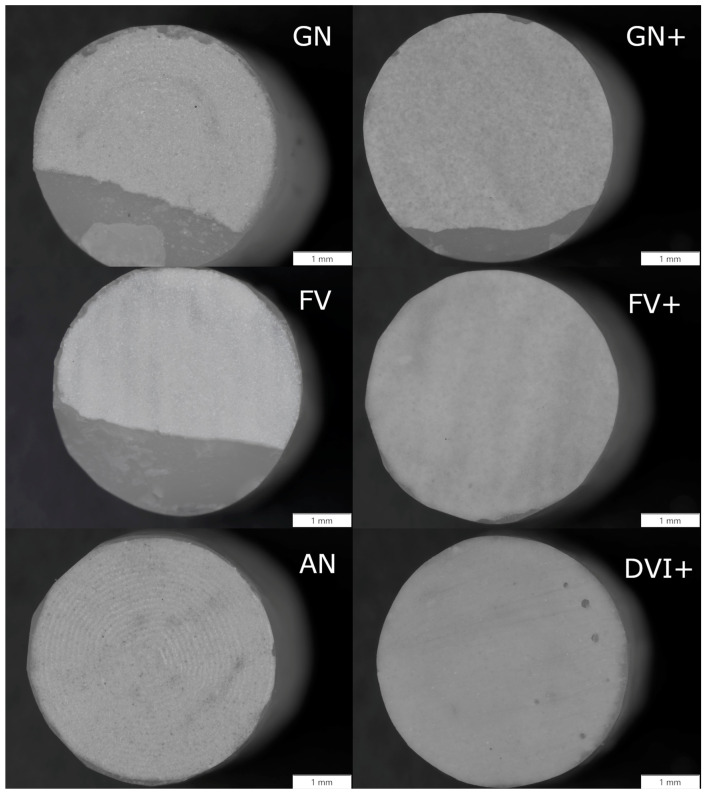
Stereomicroscope images of groups GN, GN+, FV, FV+, AN and DVI+. Specimens which achieved higher shear bond strength (SBS) had a mixed failure (GN, GN+, FV and FV+), whereas specimens with lower SBS had an adhesive failure (AN and DVI+).

**Table 1 materials-16-03286-t001:** Used veneering composites, their abbreviation signs, components and compositions.

Veneering Composite System	Sign	Components	Composition
Visio.lign	V	Combo.lign preopaque paste light (REF Co1 × 4OPL) Combo.lign preopaque catalysator (REF: CO1 × AT) Crea.lign opaquer (REF: CLFHOP03) Crea.lign dentin (REF: CLPNDA20) Crea.lign surface cleaner (REF:43000600)	Various acrylate oligomers (urethane dimethacrylate, UDMA), silanised inorganic fillers (50 wt% opalescent ceramic fillers), catalysts and colour pigments
SR Nexco	N	SR Nexco opaque (REF: Y48592) SR Nexco dentin (REF: Z01XKL) SR Gel, (REF: 573106).	Urethane dimethacrylate (UDMA), 1,10-decanediol dimethacrylate and silicon dioxide (SiO_2_, 19.8 wt%) as a micro sized filler
VITA VM LC	VI	VITA VM LC preopaque, (REF: C445913) VITA VM LC opaque(REF: C443023) VITA VM LC dentin (REF: C44324) VITA VM LC Gel(LOT: 96033)	Ethylene glycol dimethacrylate (EDMA), triethylene glycol dimethacrylate (TEGDMA) and 2-(dimethylamino)ethyl methacrylate (DMAEMA) with the micro sized filler SiO_2_ (45–48 wt%)

**Table 2 materials-16-03286-t002:** Different surface treatments of BioHPP and BioHPP plus and marks used.

Mark	Surface Treatments
A	Untreated
B	Sandblasted with 110 μm Al_2_O_3_
C	Sandblasted with 110 μm Al_2_O_3_, cleaned ultrasonically with 80% ethanol, 5 min
D	Sandblasted with 110 μm Al_2_O_3_, application of MKZ primer
E	Sandblasted with 110 μm Al_2_O_3_, application of MKZ primer and Visio.link adhesive
F	Sandblasted with 110 μm Al_2_O_3_, application of Visio.link adhesive
G	Sandblasted with 110 μm Al_2_O_3_, cleaned ultrasonically with 80% ethanol, 5 min, application of Visio.link adhesive

**Table 3 materials-16-03286-t003:** Designation of BioHPP and BioHPP plus specimens with three veneering composites (V—Visio-lign, N—SR Nexco and VI—VITA VM LC) for SBS test measurement.

PEEK Composite	Veneering Composite Used	Surface Treatment Marks						
		A	B	C	D	E	F	G
BioHPP	Visio.lign	AV	BV	CV	DV	EV	FV	GV
	SR Nexco	AN	BN	CN	DN	EN	FN	GN
	VITA VM LC	AVI	BVI	CVI	DVI	EVI	FVI	GVI
BioHPP plus	Visio.lign	/	BV+	CV+	DV+	EV+	FV+	GV+
	SR Nexco	/	BN+	CN+	DN+	EN+	FN+	GN+
	VITA VM LC	/	BVI+	CVI+	DVI+	EVI+	FVI+	GVI+

**Table 4 materials-16-03286-t004:** The thermal transitions and enthalpy of fusion of the PEEK composites determined by DSC.

PEEK Composites	*T_g_* (°C)	*T_cc_* (°C)	*T_c_* (°C)	*T_m_* (°C)	Δ*H_m_* (J/g)	Degree of Crystallinity *X_c_* (%)
BioHPP (DSC)	151.3 ± 1.3	/	287.6 ± 0.2	338.7 ± 0.2	29.9 ± 1.9	29.0 ± 1.4
BioHPP (manufacturer)	155.8 ± 2.0	/	/	343.2 ± 0.3	/	/
BioHPP plus (DSC)	150.4 ± 0.4	170.2 ± 0.6	293.0 ± 0.7	339.4 ± 0.6	25.5 ± 0.6	27.9 ± 0.7
BioHPP plus (manufacturer)	156.8 ± 2.0	/	/	345.1 ± 0.3	/	/
BioHPP (DMA)	169 ± 0.4	/	/	/	/	/
BioHPP plus (DMA)	167 ± 0.25	/	/	/	/	/

**Table 5 materials-16-03286-t005:** The viscoelastic properties of PEEK composites determined by DMA.

PEEK Composites	*E*′ at 30 °C (GPa)	*E*″ at 30 °C (GPa)	tan δ
BioHPP	4.258 ± 0.093	0.248 ± 0.004	0.153 ± 0.006
BioHPP plus	4.193 ± 0.09	0.314 ± 0.016	0.154 ± 0.001

**Table 6 materials-16-03286-t006:** Contact angles of BioHPP and BioHPP plus specimens.

Treatment of Specimen Surface	Contact Angles of BioHPP Surface		Contact Angles of BioHPP Plus Surface	
	Dist. Water (°)	Diiodomethane (°)	Dist. Water (°)	Diiodomethane (°)
Untreated	75.57 ± 2.49	56.76 ± 2.58	102.52 ± 1.26	0 ± 0
Sandblasted	89.75 ± 1.30	13.93 ± 0.67	106.87 ± 3.02	10.17 ±1.82
Polished	102.90 ± 0.70	40.80 ± 2.34	89.10 ± 1.81	40.40 ± 1.13

**Table 7 materials-16-03286-t007:** Free surface energy and its disperse and polar parts of BioHPP and BioHPP plus.

Treatment of Specimen Surface	Surface Free Energy (mN/m)		Disperse Part (mN/m)		Polar Part (mN/m)	
	BioHPP	BioHPP Plus	BioHPP	BioHPP Plus	BioHPP	BioHPP Plus
Untreated	34.10	55.51	23.67	55.06	10.43	0.45
Sandblasted	49.39	56.68	48.90	55.56	0.48	1.12
Polished	40.93	38.80	40.93	37.09	0.01	1.71

**Table 8 materials-16-03286-t008:** Roughness of BioHPP and BioHPP plus surfaces.

Treatment of Specimen Surface	BioHPP (μm)	BioHPP Plus (μm)
Untreated	0.568 ± 0.260	1.841 ± 0.382
Sandblasted	1.154 ± 0.185	1.29 ± 0.180
Polished	0.014 ± 0.002	0.02 ± 0.004

**Table 9 materials-16-03286-t009:** Shear bond strength of BioHPP and BioHPP plus specimens with various surface treatments and three different veneering composites: Visio.lign (V), SR Nexco (N) and VITA VM LC (VI). A detailed description of individual groups is given in Table 2 and Table 3.

	SBS of BioHPP (MPa)			SBS of BioHPP Plus (MPa)		
Veneering Composites						
	Visio Lign (V)	SR Nexco (N)	VITA VM LC (VI)	Visio Lign (V+)	SR Nexco (N+)	VITA VM LC (VI+)
A	7.27 ± 3.86	1.07 ± 1.19	5.39 ± 3.17	/	/	/
B	17.68 ± 3.75	5.93 ± 3.58	11.06 ± 4.60	14.13 ± 3.23	12.18 ± 2.29	10.96 ± 1.53
C	23.42 ± 2.49	14.18 ± 3.28	13.95 ± 2.55	15.57 ± 2.64	14.02 ± 2.81	6.16 ± 1.73
D	25.59 ± 3.17	9.05 ± 7.64	13.12 ± 5.18	12.44 ± 2.84	8.97 ± 1.69	2.92 ± 0.43
E	23.66 ± 3.85	19.37 ± 8.06	25.51 ± 1.94	15.70 ± 4.49	14.34 ± 3.01	7.35 ± 0.90
F	26.31 ± 4.17	19.76 ± 4.98	22.70 ± 7.87	19.62 ± 4.26	18.44 ± 3.55	15.30 ± 3.72
G	20.31 ± 6.56	25.00 ± 6.99	26.28 ± 2.94	21.53 ± 3.24	23.39 ± 2.80	14.13 ± 4.05

**Table 10 materials-16-03286-t010:** Types of failures (adhesive or mixed) of all groups, depending on the surface pretreatment and veneering composites used on BioHPP and BioHPP plus.

Type of Failure	Groups
Adhesive	AV, AN, BN; DN, FN, AVI, BVI, FVI, DV+, FV+, GV+, BN+, DN+, BVI+, CVI+, DVI+, EVI+ and FVI+
Mixed	BV, CV, DV, EV, FV, GV, CN, EN, GN, CVI, DVI, EVI, GVI, BV+, CV+, EV+, CN+, EN+, FN+, GN+ and GVI+

## Data Availability

Data sharing not applicable.

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
