# Peer review of "Determination of Shear Bond Strength between PEEK Composites and Veneering Composites for the Production of Dental Restorations"

_materials, 2023, doi:10.3390/ma16093286_

Round 1

Reviewer 1 Report

The authors of the manuscript "Determination of Shear Bond Strength between PEEK Composites and Veneering Composites for the Production of Dental Restorations" investigated the bond strength of PEEK to composite as a veneering option. The manuscript is well written but needs a revision of the methods.

Major points:

- it is unclear why three different composites were used. It should be made clear how they differ and why, for example, different results are expected from composites containing UDMA or EDMA. The hypotheses could be supported with references.

- the authors should include additional experiments after artificial aging by thermocycling. Thermocycling has been established as the standard method for artificial aging. Most of the studies (also on PEEK) show a significant reduction of the bond strength values, which have a great influence on the clinical usability.

Minor points:

- In lines 15 and 17 as well as 114 and 116, the abbreviations DSC and DMA should be explained.

- In lines 38-41, it is not clear which references the GPa values mentioned refer to.

- Is the measurement method described from line 241 according to an ISO standard?

Author Response

Dear reviewer,

  • We explained the use of three different veneering composites in lines 105 to 114 of the revised manuscript. In lines 73 we explained the selection of PEEK composites BioHPP and BioHPP plus.
  • Regarding the performance of thermomechanical tests, we contacted prof. dr. Vojkan Lazić from the Faculty of Dentistry (Belgrade, Serbia). Since the performance of thermomechanical tests is demanding and time-consuming, we will carry them out as part of further research work.
  • Minor points in the manuscript in lines 15 and 17 as well in lines 114 and 116 were corrected (lines 15, 17, 116 and 120 in the revised manuscript).
  • The references regarding the mentioned GPa values were aded in lines 37 to 40.
  • Shear bond strength was determined using a universal testing machine and custum made fixture tool (special mould – fig. 1) that enables application of uniaxial shear load exactly at the interface of PEEK/Veneering composite according to the standard ISO 10477 for determination of the bond strength of polymer based crown and bridge materials (line 247).

Reviewer 2 Report

I suggest the authors to highlight the importance of studying peek composites, preferably in the Introduction section. Why and when should it be a TiO2 or titanium alloy (such as Ti-6Al-4V) substitute in dentistry? For exemple: In line 50 - why they are not suitable for aesthetic restorations? Are titanium implants also indicated for anterior restorations?

I also propose the authors to explain why they chose these two peek composites, as well as the three used veneering composites (in the Introduction or Material and Methods).

Finally, I suggest the authors to review the tables and figures editing rules.

Author Response

Dear reviewer,

  • In lines 39 to 46 we explained why there was a need to use a material that would have better properties than titanium implants and why PEEK and its composites could be suitable materials.
  • In lines 55 to 58 we mention the drawback of PEEK and PEEK composites which, when used alone, are not able to achieve the needed aesthetics of dental restorations.
  • The selection of BioHPP and BioHPP plus and three veneering composites as testing materials for our study is described in lines 73 and 104 to 113 of the revised manuscript. BioHPP and BioHPP plus are alternative materials when patients want a non-metalic dental restauration because of their personal statements or possible allergies to titanium or its alloys.

Round 2

Reviewer 1 Report

This revision of the manuscript "Determination of Shear Bond Strength between PEEK Composites and Veneering Composites for the Production of Dental Restorations" contains improvements to the original version.

However, no further thermocycling tests were performed, which severely limits the clinical implications of the study. I recommend to perform the additional tests. In previous studies on peek, the SBS declined almost to zero after thermocycling. 

The conclusion drawn from line 555 is not supported by the data. The limitation due to the lack of thermocycling in the experimental design needs to be identified.

Minior points:

The information on the selection of materials starting at line 104 is helpful but should rather be moved to the introduction section.

Author Response

Dear Reviewer 1,

please accept our thanks for your time and for considering this submission. We would like to thank you for providing both the supporting and constructive comments. Following all suggestions made by the Reviewers, we revised the manuscript accordingly. In the attached ‘Response to the Reviewers’ file, detailed answers to each point, including the corresponding changes in the manuscript, are given. The changes in the revised manuscript are marked in red.

We respect the Reviewer’s opinion on the proposed thermocycling test, however, we explain why we cannot follow this advice at the moment. We believe that the information on thermocycling test is currently beyond the scope of this paper, in which we focus on the in vitro and surface properties of the PEEK-polymer composites.

Regarding the proposed thermocycling test, the time of seven days, given for corrections of the manuscript, is not enough to perform such a test. Also we believe that thermocycling would be a test for further research. Namely, new specimens for all groups (around 100 specimens) have to be prepared. The time needed to prepare specimens according to the CAD/CAM technique is approx. 30 specimens a day and the same for the pressed specimens. The manual application of three different veneering composites takes at least half an hour for one specimen. Also, we need to make arrangements for experimental work with four dental laboratories, where we made specimens for the present study. The thermocycling test of a minimum 91 specimens, which need to be produced for this study, would take a minimum of 83 hours, taking into consideration the measurements of 5000 cycles at 5 °C and 55 °C. We will carry these tests out as a part of further research work. As mentioned, we contacted regarding thermocycling prof. dr. Vojkan Lazić from the Faculty of Dentistry (Belgrade, Serbia).

Regarding the conclusion in line 555, we agree with the Reviewer’s opinion, that we did not confirm our statement.

We hope that the revised manuscript meets the Editor’s and the Reviewers’ requirements.

Best regards,